# Enhanced Fluorescence Characteristics of SrAl_2_O_4_: Eu^2+^, Dy^3+^ Phosphor by Co-Doping Gd^3+^ and Anti-Counterfeiting Application

**DOI:** 10.3390/nano13142034

**Published:** 2023-07-09

**Authors:** Peng Gao, Quanxiao Liu, Jiao Wu, Jun Jing, Wenguan Zhang, Junying Zhang, Tao Jiang, Jigang Wang, Yuansheng Qi, Zhenjun Li

**Affiliations:** 1Beijing Key Laboratory of Printing and Packaging Materials and Technology, Beijing Institute of Graphic Communication, Beijing 102600, China; gaopeng08252022@163.com (P.G.); drllqx@163.com (Q.L.); wujiao20220912@163.com (J.W.); jj20021017@163.com (J.J.); zhangwenguan@bigc.edu.cn (W.Z.); 2School of Physics, Beihang University, Beijing 100191, China; zjy@buaa.edu.cn; 3CAS Center for Excellence in Nanoscience, Beijing Key Laboratory of Micro-Nano Energy and Sensor, Beijing Institute of Nanoenergy and Nanosystems, Chinese Academy of Sciences, Beijing 101400, China; jiangtao@binn.cas.cn; 4National Center for Nanoscience and Technology, CAS Key Laboratory of Nanophotonic Materials and Devices (Preparatory), Beijing 100190, China; 5The GBA Research Innovation Institute for Nanotechnology, Guangzhou 510700, China

**Keywords:** SrAl_2_O_4_, Eu^2+^, Dy^3+^, Gd^3+^, rare-earth elements, bandgap, long afterglow, mechanism of afterglow, anti-counterfeiting

## Abstract

A series of long-afterglow luminescent materials (SrAl_2_O_4_: Eu^2+^ (SAOE), SrAl_2_O_4_: Eu^2+^, Dy^3+^ (SAOED) and SrAl_2_O_4_: Eu^2+^, Dy^3+^, Gd^3+^ (SAOEDG)) was synthesized via the combustion method. Temperature and concentration control experiments were conducted on these materials to determine the optimal reaction temperature and ion doping concentration for each sample. The crystal structure and luminescent properties were analyzed via X-ray diffraction (XRD), photoluminescence (PL), and afterglow attenuation curves. The outcomes demonstrate that the kind of crystal structure and the location of the emission peak were unaffected by the addition of ions. The addition of Eu^2+^ to the matrix’s lattice caused a broad green emission with a central wavelength of 508 nm, which was attributed to the characteristic 4f^6^5d^1^ to 4f^7^ electronic dipole, which allowed the transition of Eu^2+^ ions. While acting as sensitizers, Dy^3+^ and Gd^3+^ could produce holes to create a trap energy level, which served as an electron trap center to catch some of the electrons produced by the excitation of Eu^2+^ but did not itself emit light. After excitation ceased, this allowed them to gently transition to the ground state to produce long-afterglow luminescence. It was observed that with the addition of sensitizer ions, the luminous intensity of the sample increased, and the afterglow duration lengthened. The elemental structure and valence states of the doped ions were determined with an X-ray photoelectron spectrometer (XPS). Scanning electron microscopy (SEM) and energy dispersive X-ray spectroscopy (EDX) were used to characterize the samples. The results show that the sample was synthesized successfully, and the type and content of ions in the fluorescent powder could be determined. The fluorescence lifetime, quantum yield, bandgap value, afterglow decay time, and coordinate position in the coherent infrared energy (CIE) diagram of the three best sample groups were then analyzed and compared. Combining the prepared phosphor with ink provides a new idea and method for the field of anti-counterfeiting through screen printing.

## 1. Introduction

Phosphors have been around for a long time. SrAl_2_O_4_ is a phosphorescent material with broad application prospects. It is a phosphorescent material prepared via sintering SrCO_3_ and Al_2_O_3_ at high temperatures. It has good fluorescence performance and long afterglow characteristics. Its fluorescence peak is located in the green area, and the fluorescence lasts for several hours [1,2,3]. A phosphorescent material is a substance that can emit light of a specific wavelength after being excited. SrAl_2_O_4_ was first discovered by Japanese scientists in 1996 and classified as a new type of long-lasting luminescent material. A long afterglow refers to the ability of a material to remain glowing for hours or even days after limited energy input. The reason why SrAl_2_O_4_ has this unique luminescent property is due to its special crystal structure and material composition. In the past few decades, the research on SrAl_2_O_4_ has gradually deepened and has attracted the attention of academia and industry. Researchers have used different chemical methods to improve the synthesis and composition of the material [4,5], further improving its performance and application range. For example, pure SrAl_2_O_4_ materials can be synthesized via the sol–gel and solid-phase methods [6]. Meanwhile, some studies have also considered the composite doping of multiple rare earth element ions in SrAl_2_O_4_. By doping rare earth element ions such as europium and dysprosium, the luminescence characteristics of SrAl_2_O_4_ can be regulated, and different Eu ion doping agents also have an impact on the luminescence intensity and color [7,8,9]. In addition, by changing the crystal structure and doping ion concentration of SrAl_2_O_4_, its luminescence characteristics can be regulated to meet the needs of different applications [10]. At present, SrAl_2_O_4_ has become an important material in the fields of fluorescent materials, marker materials, biosensors, displays, and other fields. At the same time, combined with the development of artificial intelligence and other technologies and has broad application prospects [11,12,13,14,15].

The spectral characteristics of phosphors are mainly determined by the selected dopant ions [16]. Doping with rare-earth ions is a method used for changing the characteristics of materials. Inorganic nanomaterials’ crystallographic phases, morphologies, sizes, and electrical configurations can be altered by doping them with rare-earth ions. By doping with appropriate rare-earth elements, the material’s performance can be optimized, such as increasing the luminous efficiency of the material, expanding its luminous range, or enhancing its spectral stability [17]. In 2008, Song et al. [18] prepared SAOEDG fluorescent powder using a combustion method, adding a small amount of H_3_BO_3_ as a flux and stirred the prepared solution at 70 °C for 4 h. This helped the solution to fully reflect in a muffle furnace. However, this method was not used in this experiment.

In recent years, counterfeiting and poor-quality goods have been repeatedly prohibited, and the significance of anti-counterfeiting technology is becoming increasingly important. At present, anti-counterfeiting technology mainly includes anti-counterfeiting paper technology, anti-counterfeiting ink technology, secure printing technology, etc. Among them, anti-counterfeiting ink technology is an extremely important field in anti-counterfeiting technology, and UV-excited fluorescent ink is a key application therein [19,20].

In this study, green-emitting phosphors SAOE, SAOED, and SAOEDG were successfully prepared via the combustion method. We studied and compared their crystal structures, chemical states, composition, fluorescence lifetime, quantum yields, and afterglow mechanisms. Their bandgap values were determined through theoretical and experimental calculations. Then, the coordinate positions of the three samples in the Commission Internationale de I’Eclairage chart were compared. Finally, the fluorescent powder and ink were mixed to form anti-counterfeiting fluorescent ink, and anti-counterfeiting applications were studied via screen printing.

## 2. Materials and Equipment

### 2.1. Materials and Synthesis

According to the molecular formula Sr_1−x−y−z_Al_2_O_4_: Eu_x_^2+^, Dyy3+, Gdz3+, a series of SrAl_2_O_4_: Eu^2+^, SrAl_2_O_4_: Eu^2+^, Dy^3+^, and SrAl_2_O_4_: Eu^2+^, Dy^3+^, Gd^3+^ phosphors was successfully prepared via by a simple combustion method at different temperatures. SrCO_3_(A.R.), Al_2_O_3_(A.R.), Eu_2_O_3_(A.R.), Dy_2_O_3_(A.R.), Gd_2_O_3_(A.R.), and urea (CO(NH_2_)_2_) were used as raw materials; they were purchased from Tianjin Chemical Reagent Factory. Deionized water was self-made. Without additional purification, all reagents were utilized right away after receiving them.

At first, SrCO_3_, Al_2_O_3_, Eu_2_O_3_, Dy_2_O_3_, and Gd_2_O_3_ were separately dissolved in nitric acid to obtain transparent and clear solutions of Sr(NO_3_)_2_ (0.5 mmol/mL), Al(NO_3_)_3_ (1 mmol/mL), Eu(NO_3_)_3_ (0.1 mmol/mL), Dy(NO_3_)_3_ (0.1 mmol/mL), and Gd(NO_3_)_3_ (1 mmol/mL). According to the stoichiometric ratio, we weighed the correct solutions of Sr(NO_3_)_2_, 4 mL of Al(NO_3_)_3_ • 9H_2_O, Eu(NO_3_)_2_, Dy(NO_3_)_3_, Gd(NO_3_)_3_, 2.2 g of urea, and an appropriate amount of deionized water into a crucible, which we mixed thoroughly until the solution was clear and transparent. Then, the solution was burned in a muffle furnace for 3–5 min, until we heard a "bang" and observed the flame. We then took out the product after a moment of cooling. After the combustion reaction, a milky white, mushroom-like substance was produced. After cooling and grinding, green SrAl_2_O_4_ luminous powders were produced.

The Eu^2+^-doped samples were prepared via this experimental operation: the doping amount of Eu(NO_3_)_3_(0.1 mmol/mL) was 0.5%, 2%, 4%, 6%, 8%, or 10%. The Eu^2+^ and Dy^3+^ co-doped samples were prepared via the same experimental procedure. The doping amount of 2%Eu^2+^ and Dy(NO_3_)_3_ (0.1 mmol/mL) was 0.1%, 0.5%, 1%, 2%, 4%, or 6%. The Eu^2+^, Dy^3+^, and Gd^3+^ co-doped samples were prepared via the same experimental procedure; the doping amount of 2%Eu^2+^, 0.5%Dy^3+^, and Gd(NO_3_)_3_ (1 mmol/mL) was 0.1%, 1%, 2%, 4%, 6%, or 8%. Afterward, using the same experimental method, the corresponding samples were prepared using ions of the same concentration at a reaction temperature of 500 °C, 600 °C, 700 °C, 800 °C, or 900 °C. Finally, identify the optimal reaction temperature and ion doping concentration for the three fluorescent powders were determined, Afterward, we determined their optical properties and compared them.

### 2.2. Instruments

The crystal structures of the synthesized samples were recorded ranging from 10° to 90° on a powder X-ray diffractometer (D/max 2200PC by Rigaku) with Cu Kα radiation (λ = 1.54 Å). We used a monochromatic Al-Kα (hv = 1486.6 eV) X-ray source and the charge was corrected by polluting carbon C1s = 284.8 eV for X-ray photoelectron spectroscopy (TAmerican Thermo) for the analysis of the elemental and chemical state information of the samples. Photoluminescence spectroscopy (PL) and photoluminescence excitation spectroscopy (PLE) were performed with a fluorescence spectrometer (F4700, Hitachi, Japan). We observed and compared the morphology of the fluorescence with a scanning electron microscope (Quanta 250 FEG, Hitachi, Japan). The elemental compositions and contents were measured with a scanning electron microscope energy meter (SU8020, Hitachi, Japan). The UV diffuse reflectance absorption spectra were measured with a UV-3600 Ultraviolet–Visible Near-Infrared Spectrophotometer (Shimadzu Corporation, Kyoto, Japan). The afterglow attenuation curve, fluorescence lifetime, and quantum yield of the phosphors were measured with a transient steady state fluorescence spectrometer (FLS1000, Edinburgh, UK).

### 2.3. Band Structure and Density of States of SrAl_2_O_4_

The calculations of the band structure and density of states of SrAl_2_O_4_ were performed using density functional theory (DFT). All DFT calculations were carried out using the Vienna ab initio Simulation Package (VASP) [21]. The Perdew–Burke–Ernzerhof (PBE) [22] exchange–correlation functional and projector augmented wave (PAW) [23] pseudopotential were adopted with spin polarization. During the structure optimization, the convergence criterion of the total energy was set to 10${}^{-6}$ eV, and the atoms were relaxed until the force acting on each atom was less than 0.01 eV/Å. Gaussian smearing of 0.05 eV to the orbital occupation was applied. A plane-wave cut-off energy of 500 eV was used in all computations. The Brillouin-zone integrations were conducted using Monkhorst–Pack (MP) grids of special points with the separation of 0.06 Å${}^{-1}$.

### 2.4. Preparation of Ink

Ink was prepared with solvent I (32%), solvent II (36%), polyamide resin (32%), wax powder (auxiliary), and an appropriate amount of glass beads. The matrix of glass beads was made of silicate glass. This material usually has high visible light transparency to ensure that the fluorescence effect can be fully displayed. Solvent I was n-butanol, and solvent II was composed of isopropanol and methylcyclohexane in a ratio of 1:1. We put them in tin cans, sealed them, placed them in a shaker to disperse for 25 to 35 min, and filtered them. Then, we added an appropriate amount of fluorescent powder and stirred them well to form a fluorescent ink.

## 3. Results and Discussion

### 3.1. Phase Analysis

The phase composition and purity of SrAl_2_O_4_ phosphors doped with different rare-earth ions were investigated using X-ray powder diffraction. Figure 1a shows the X-ray diffraction (XRD) pattern of the samples prepared under different ion doping conditions. SrAl_2_O_4_ had two phases: a high-temperature hexagonal phase (β-phase) and a low-temperature monoclinic phase (α-phase) [18,24]. The transition temperature occurred at 650 °C. SrAl_2_O_4_ was a stable compound in a SrO–Al_2_O_3_ system. It had a stable monoclinic phase at room temperature, which transformed into hexagonal when heating at temperatures above 650 °C, which returned to monoclinic at the same temperature during cooling [25]. In Figure 1a, the XRD profile exhibits distinct peaks at 2θ values 19.951°, 28.386°, 29.275°, 29.922°, and 35.113°. It was found that these peaks match well with the data registered in the Joint Committee on Powder Diffraction Standards (JCPDS) data file (JCPDS card number 34–0379) for monoclinic SrAl_2_O_4_. Pure monoclinic phase (space group $P2_{1}$) diffraction peaks of SrAl_2_O_4_ were predominant in the XRD pattern. Its lattice constants were a = 8.442 Å, b = 8.822 Å, c = 5.161 Å, and β = 93.415° [18,26,27]. Its crystal structure is shown in Figure 1b. Additionally, no other products or starting materials were observed, implying that the small amount of doped rare-earth ions had almost no effect on the SrAl_2_O_4_ phase composition [28]. General Structure Analysis System (GSAS) software was used to obtain the Rietveld refinement XRD patterns of the SAOE, SAOED, and SAOEDG samples to further confirm the phase purity; as shown in Figure 1c–e, there were no additional impurity peaks between the observed patterns (black cross) and the calculated data (red line). For the SAOE sample, its refinement parameter values were $R_{wp}$ = 6.17%, $R_{p}$ = 5.94%, and $\chi^{2}$ = 7.58; for the SAOED sample, its refinement parameter values were $R_{wp}$ = 5.94%, $R_{p}$ = 4.38%, and $\chi^{2}$ = 5.10; for the SAOEDG sample, its refinement parameter values were $R_{wp}$ = 9.78%, $R_{p}$ = 6.02%, and $\chi^{2}$ = 6.47. It is generally believed that the values of $R_{wp}$ and $R_{p}$ are less than 10%, indicating a good finishing effect. This demonstrated the single-phase nature of the SrAl_2_O_4_ host [24,25,26,27,28].

The positions of the ionic radius of the different dopants and co-dopants (sensitizers) in the SrAl_2_O_4_ matrix are different. The ionic radius of Sr^2+^ is 1.21 Å, the ionic radius of Eu^2+^ is 1.20 Å, the ionic radius of Dy^3+^ is 0.97 Å, and the ionic radius of Gd^3+^ is 1.00 Å [25,29]. In the crystal structure of SrAl_2_O_4_, the Sr site is typically found in a six-coordinated octahedral coordination environment. However, under certain conditions, the rare-earth elements Dy and Gd can indeed replace Sr at this site. Due to the slightly unique chemical properties of Dy and Gd in high-valence rare-earth elements, the feasibility of replacing the Sr in SrAl_2_O_4_ may be limited. For example, if a Dy^3+^ replaces a Sr^2+^ ion, there would be $1^{+}$ charge incompatibility. It is worth noting that Eu, with its fewer possible valence states and better coordination properties, is generally more readily able to substitute for Sr. For SrAl_2_O_4_: Eu^2+^, RE3+(RE = Dy, Gd), it only affects the duration and intensity of the afterglow, making the duration and intensity of the afterglow longer and stronger. Moreover, the radii percentage deviation between the Eu^2+^ ions and Sr^2+^ ions should be less than 30%. The radii difference percentage can be estimated with the following formula [30]:(1)$$D_{r}=100\%\times \frac{R_{m}\left(CN\right)-R_{d}\left(CN\right)}{R_{m}\left(CN\right)}$$
where R_*m*_ (CN) and Rd (CN) refer to the radii of the Sr^2+^ and Eu^2+^ ions. The values of D_r_ between Sr^2+^ and Eu^2+^ were calculated as 0.8%. Consequently, it can be reasonably stated that the Eu^2+^ ions replaced the Sr^2+^ ions in SrAl_2_O_4_. In contrast, Eu^2+^ did not fit at all into the small Al^3+^ site [31]. This proves that Eu^2+^, Dy^3+^, and Gd^3+^ ions were successfully doped into the SrAl_2_O_4_ matrix. Eu was the luminescent center, and Dy and Gd were sensitizers. According to the XRD pattern, the crystallite sizes (D) of each sample could be calculated. Generally, the grain size and half height width of the sample can be calculated by Scherrer’s formula (a famous formula for XRD analysis of grain size) [32]:(2)$$D=\frac{k\lambda }{\beta cos\theta }$$
where *D* is the crystallite size (nm), *K* is 0.89 (Scherrer constant), λ is 0.15406 nm (wavelength of the X-ray source), β is the FWHM (radians), θ is the peak position (radians). In the origin software, the corresponding values of β and θ can be obtained by multi-peak fitting of the XRD spectrum of each sample through the Gaussian formula. After substituting into the formula, the average grain sizes of SAOE, SAOED, and SAOEDG were approximately 28.284, 31.524, and 29.887, respectively. Detailed data can be found in Table 1, Table 2 and Table 3. In the combustion method in this experiment, H_3_BO_3_ was not added as a flux, and there was no long-term stirring at low temperature, leading to failure to better promote the reaction during combustion [18]. This may be the reason for its slightly smaller grain size.

### 3.2. Elemental Analysis

Since the luminescence performance of the doped elements was significantly influenced by their chemical valence, it was necessary to determine the chemical valence of the dopant ions in the prepared samples. The chemical composition and oxidation states of the various elements (mainly europium) in the SrAl_2_O_4_: Eu^2+^, Dy^3+^, Gd^3+^ phosphor were analyzed using X-ray photoelectron spectroscopy (XPS), as shown in Figure 2. It shows the outcomes of this phosphor’s survey and high-resolution elemental scans. As shown in Figure 2a, XPS the shows prominent peaks corresponding to Sr 3d, Al 2p, O 1s, C 1s, Eu 3d, Dy 3d, and Gd 4d. The high-resolution XPS spectra of Sr 3d are shown in Figure 2b, where it can be seen that the Sr 3d peak represents two peaks. These are the Sr 3d_5/2_ (133.71 eV) and Sr 3d_3/2_ (135.54 eV) peaks, which are associated with the Sr from the two SrAl_2_O_4_ sites [33]. The Al 2p and O 1s signals were located at 74.38 eV and 531.73 eV, respectively. As shown in Figure 2e, Eu 3d can be observed in the spectrum at the expected position, and europium is at the divalent oxidation state; the peak centers are at 1125.85 eV and 1154.31 eV, which belong to Eu^2+^ 3d_5/2_ and Eu^2+^ 3d_3/2_. The 3d_5/2_ and 3d_3/2_ peaks would be located close to 1134 eV and 1165 eV, respectively, if europium was present in its trivalent state [34]. The Gd peaks in Figure 2f were found at 143.87 eV for 4d_5/2_, indicating that the Gd of the nanoparticles existed in the form of oxide and was positive trivalent. Due to the extremely low content of Dy ions (only 0.5%), it remained at the noise level, and its high-resolution picture is not shown here. But, in the XPS spectrum, the binding energy of Dy could be detected as Dy 3d 1299.31 eV. All of the elements in SrAl_2_O_4_: Eu^2+^, Dy^3+^, Gd^3+^ were in the expected valence state, according to XPS. All of these elements’ peaks were consistent with their respective standard central values [35,36].

### 3.3. SEM Analysis

SEM images display the granular microstructure of luminescent materials. Figure 3a–c show the SEM images of three fluorescent powders: SAOE (2% Eu, 600 °C), SAOED (2% Eu, 0.5% Dy, 800 °C), and SAOEDG (2% Eu, 0.5% Dy, 2% Gd, 600 °C). The surface of the powder samples was discovered to have numerous pores and spaces created by escaping gases after combustion. When a gas escapes under high pressure, pores are formed with the formation of small particles near the pores. The samples’ microstructures exhibited the inherent characteristics of the combustion process. The irregular and non-uniform forms of the particles, as depicted, could be linked to the combustion flame’s uneven temperature and mass flow distribution [37]. Figure 3d–l show the elemental composition, the contents of the three SrAl_2_O_4_ phosphors, and the elemental mapping images. The necessary elements O, Al, Sr, Eu, Dy, and Gd were recorded in the EDX spectra. Their contents were almost consistent with the formula of the material. It is evident that due to the influence of oxygen in the air during the tested operations, the atomic percentages of the Eu^2+^ ions were measured to be approximately 0.97%, 1.95%, and 1.58%; the atomic percentages of the Dy^3+^ ions were measured to be approximately 0.42% and 0.32%; and the atomic percentage of the Gd^3+^ ions was measured to be approximately 1.82%, which is a little lower than the standard value of 2%Eu, 0.5%Dy, and 2%Gd based on the chemical formula. The element color mapping images show that Eu, Dy, and Gd were evenly distributed on the particles, further suggesting that Eu^2+^, Dy^3+^, and Gd^3+^ ions were successfully incorporated into the SrAl_2_O_4_ nanoparticles.

### 3.4. Optical Characterization

Figure 4a shows the fluorescence emission spectra of the SAOE (2% Eu, 600 °C), SAOED (2% Eu, 0.5% Dy, 800 °C), and SAOEDG (2% Eu, 0.5% Dy, 2% Gd, 600 °C) phosphors under 365 nm ultraviolet excitation. It can be seen that the maximum emission peaks of the three phosphors are all at 508 nm. Figure 4b depicts the excitation spectrum following excitation, with the emission peak value of 508 nm taken from the emission spectrum. The peak wavelength is 365 nm, which is compatible with the emission spectra, showing that 365 nm is the major excitation wavelength of SAOE, SAOED, and SAOEDG.

It is well known that in SrAl_2_O_4_: Eu^2+^, RE3+ (RE = Dy, Gd) phosphors, Eu^2+^ ions are the luminous centers and that the transition of Eu^2+^ ions from the high-energy state 4f^6^5d^1^ to the low-energy state 4f^7^ is what causes the photo-excited luminescence. It is worth noting that these emission spectra all have broad bands, which could be due to the following factor: the 4f^7^ electronic configuration may be blended with the excited state 4f^6^5d^1^ of Eu^2+^ ions according to crystal field and ligand theory [38,39]. Further evidence that the divalent europium ions of the residual rare-earth ions still exist in the grain boundaries is provided by the absence of emission peaks in the red area between the energy levels in the 4f sublayer of Eu^3+^. Gadolinium and dysprosium trivalent ions can create both shallow and deep traps in SrAl_2_O_4_, whereas Eu^2+^ ions can only form shallow traps. Dy^3+^ ions and Gd^3+^ ions function to sensitize the luminous center during the luminescence process. Dy^3+^ ions and Gd^3+^ ion as sensitizers absorb the excitation light and transfer the energy to the luminescence center Eu^2+^ to enhance its luminosity [40]. From Figure 4a, it can be seen that the phosphor doped with Dy^3+^ ions and Gd^3+^ ions in SrAl_2_O_4_: Eu^2+^ has a higher luminescence intensity, followed by the phosphor doped with Dy^3+^ ions in SrAl_2_O_4_: Eu^2+^. In the absence of any ion sensitization, the luminescence intensity of SrAl_2_O_4_: Eu^2+^ is relatively the lowest.

The combustion reaction temperature and ion doping concentration are very important for fluorescent powders, as they directly affect the structure and luminescent properties of the sample. Therefore, we prepared multiple sets of samples to determine the optimal reaction temperature and ion doping concentration for SAOE, SAOED, and SAOEDG phosphors. Figure 5a shows the emission spectra of SAOE at 500 °C, 600 °C, 700 °C, 800 °C, and 900 °C. The spectral characteristics at the same wavelength excitation did not significantly alter when the reaction temperatures were different. The spectrum shows a green wide emission band centered at 508 nm. When the reaction temperature was 600 °C, the luminescence intensity of the sample was optimal. According to Figure 5b,c the optimal reaction temperatures for SAOED and SAOEDG are 800 °C and 600 °C. According to Figure 5d–f, the optimal ion doping concentrations are 2%Eu, 0.5%Dy, and 2%Gd. Afterward, we used the samples produced with the optimal temperature and concentration to research and compare their optical properties.

The term “fluorescence lifetime” describes the amount of time that passes after the excitation source has been turned off before the fluorescence intensity drops to 1/*e* of the original value. Under a xenon lamp, the fluorescence lifetime curves of the SAOE, SAOED, and SAOEDG samples were measured. Figure 6 illustrates the well-fitted curve. The two regimes of the fluorescence lifetime curves are an early quick decay and a subsequent gradual decay. Generally, the fluorescence lifetime decay curve can be best approximated by the following exponential relationships [41]: (3)$$I=I_{0}+A_{1}e^{-\frac{t}{\tau_{1}}}+A_{2}e^{-\frac{t}{\tau_{2}}}$$
where *I* is the phosphorescence intensity; $I_{0}$, $A_{1}$, and $A_{2}$ are constants; t is the time; $\tau_{1}$ and $\tau_{2}$ are the exponential component decay times. The values of $\tau_{1}$ and $\tau_{2}$ of the phosphors were obtained using Origin 2022 software. Formula (4) [42] can also be used to determine average lifetimes. The detailed parameters are listed in Table 4.
(4)$$\tau_{ave}=\frac{A_{1}{\tau_{1}}^{2}+A_{2}{\tau_{2}}^{2}}{A_{1}\tau_{1}+A_{2}\tau_{2}}$$

According to the formula, the average lifetimes of SAOE, SAOED, and SAOEDG were 504 ns, 696 ns, and 839 ns, respectively. It was found that when rare-earth ions Dy and Gd were added, the fluorescence lifetimes of SAOED and SAOEDG were longer than that of SAOE.

Furthermore, a crucial characteristic of a luminous material is the photoluminescence quantum yield (PLQY). The above samples were measured in the scatter range of 359.00 nm to 371.00 nm and emission range of 437.00 nm to 678.00 nm. The ratio of the photon number emitted by the sample (ε) to the photon number absorbed by the sample (α) is typically used to calculate the luminescence quantum yield (QY) of phosphors. Figure 7 shows the quantum yield spectra and calculation results of the three SAOE, SAOED, and SAOEDG samples. According to the following equation [43], the QY can be calculated: (5)$$QY=\frac{\varepsilon }{\alpha }=\frac{S_{em}}{S_{o}-S}$$
where $S_{em}$ is the quantity of photons emitted by the microspheres and is the integrated intensity of the sample’s emission light. S0 and S are the integrated intensities of the scattered light of the whiteboard and the samples, respectively. In this paper, the color red represents a whiteboard that was compared with a sample wire to determine the quantum yield. Black represents the sample line. The area difference between the emission portion (437–678 nm) and the scatter region (359–371 nm) is the quantum yield. It can be seen that the quantum yields of SAOE, SAOED, and SAOEDG were 65.87%, 33.05%, and 30.59%, respectively. The reason why the quantum yields of the SAOED phosphors and SAOEDG phosphors were lower than that of SAOE phosphors is that the addition of Dy ions and Gd ions introduced non-radiative dissipation channels, and the ions could absorb part of the energy into their energy levels; this part of the energy was not transferred to the Eu ions, resulting in a decrease in quantum yield. In addition, the addition of Dy ions and Gd ions may have also caused deformation of the lattice structure in SrAl_2_O_4_, which would also affect the efficiency of energy transfer, thereby further reducing the quantum yield. In contrast, SAOE phosphors only contains Eu^2+^ ions, the energy level structure of Eu^2+^ ions is more special, and there is a higher probability of spin inversion at the excited state energy level, which can transfer energy to the ground state energy more efficiently, thereby maintaining a high quantum yield [44,45,46].

Luminescent properties are closely related to the bandgap value. Therefore, the band structure of the SrAl_2_O_4_ host was calculated using density functional theory (DFT). As shown in Figure 8a, the valence band (VB) maximum and conduction band (CB) minimum are located at the same point, indicating that the SrAl_2_O_4_ host had a direct bandgap (Eg), with a predicted Eg value of approximately 4.31 eV. We measured the UV diffuse reflectance spectra of the obtained SAOE, SAOED, and SAOEDG to determine their bandgap values. Figure 8c–e depicts the UV–Vis diffuse reflectance spectra of synthetic samples obtained in the 200–800 nm range. According to the results of UV–Vis diffuse reflectance, the bandgap can be calculated using the Tauc equation [47]:(6)$${\left(\alpha hv\right)}^{2}=A\left(hv-E_{g}\right)$$
where *A* is a material-dependent constant, $E_{g}$ is the energy bandgap, and hv is the photon energy. The optical absorption coefficient is α. The plot of (α*hv*)^2^ against energy is shown in the inset in Figure 8, which enables extrapolation of the straight-line graph at (α*hv*)^2^ = 0 to calculate the energy gap. From these figures, it can be seen that there are bandgaps of 5.15 eV, 5.02 eV, and 5.05 eV for SAOE, SAOED, and SAOEDG compounds, respectively. The experimental value was higher than the calculated band gap of 4.31 eV, which could be explained by the well-known restrictions of density functional theory (DFT)-based approaches [48]. It can be seen that the addition of rare-earth ions had little effect on the bandgap value. This is acceptable and reasonable with the findings of the published literature [26,49,50,51]. Figure 8b displays the total and partial densities of states (DOS) of the SrAl_2_O_4_ host. The Sr 5s states, which are mixed with the Al 3s and 3p orbitals, make up the majority of the conduction band between 5.2 and 9 eV. O 2p states and a small amount of Al 3s and 3p states make up the bands between −5 and 0 eV. The assignable lower electronic bands are as follows: The Sr 4p levels form a narrow band at around −14 eV, whereas the oxygen 2s states produce a wider band with a peak at around −17 eV, with a small contribution from the Al 3s and 3p states as well. Finally, the Sr 4s states are the source of a deep band at around −32.5 eV. All bands are quite narrow (except for the conduction and upper valence bands), which suggests that the electronic states are highly localized.

Figure 9 shows the afterglow characteristics following UV-vis lamp irradiation. To guarantee consistency, all data were obtained 2 min after the excitation source was turned off. The samples’ decay process is divided into two categories: slow decay and fast decay. The quick decay processes occur first and dominate the intensity, whereas the slow decay processes occur afterward and result in the long afterglow behavior. Here, we used Formulas (3) and (4) mentioned above to fit and calculate the average afterglow time of the samples. Detailed data are in Table 5. It can be clearly seen that SAOE has almost no afterglow. The visible afterglow of SAOED can reach up to 55 s. And the initial luminescence intensity of SAOEDG is greater than those of SAOE and SAOED. The short afterglow time of this phosphor is because its energy transfer is done by non-radiative energy transfer, which is very efficient and can quickly transfer energy to the surrounding material, resulting in residual fluorescence. The energy dissipates quickly, resulting in a short afterglow. However, it has good fluorescence intensity and durability, so it is very suitable for applications in the detection and measurement of transient fluorescent signals [18,34].

It can be complicated to accurately describe afterglow decay, which is the return to the ground state of electrons with various trap depths. It is generally agreed that in SrAl_2_O_4_: Eu^2+^, RE^3+^ (RE = Dy, Gd) luminescent materials, Eu^2+^ ions act as the luminescence center, and Dy^3+^ and Gd^3+^ ions not only play the trap-level role but also creae trap levels with suitable depth, which let the phosphors possess stronger and longer-lasting luminescence [52]. Under the irradiation of a light source, Eu^2+^ ions undergo the transition 4f^6^5d^1^ to 4f^7^ levels, resulting in greenish emission. The electrons of Eu^2+^ in the 4f level transfer to the 5d level; the holes generated in the 4f valance band change Eu^2+^ to Eu^+^. Some of the accommodating holes in the valance band migrate to the conduction band and are trapped by the hole traps offered by Dy^3+^ and Gd^3+^, which increases the number of Dy^3+^ and Gd^3+^ with trapped holes from the valance band to become Dy^4+^ and Gd^4+^ [53]. This process is equivalent to the energy storage process. Meanwhile, the generated holes can travel back to the excited Eu^+^, and this turns Eu^+^to Eu^2+^, finally helping to achieve the ground state of Eu^2+^, which is balanced by emitting energy in the form of light, thereby generating luminescence in the visible region [54].

Color coordinate analysis is a crucial component in analyzing the performance of phosphors. Color coordinates are used to indicate the color of any phosphor substance in general. Hence, the color coordinates for the samples were determined using the photoluminescence (PL) emission spectra data and the chromatic standard published in 1976 by the Commission Internationale de I’Eclairage (CIE 1976). Using the CIE 1976 chromaticity diagram, the luminescence color of the samples stimulated under 365 nm was characterized (as shown in Figure 10). The CIE chromaticity coordinates x and y were calculated from the CIE tristimulus values (XYZ) by using the formula [55]:(7)x=X/X+Y+Z;y=Y/X+Y+Z

Every natural color can be identified by (*x*, *y*) coordinates. Apparently, the color coordinates of SAOE, SAOED, and SAOEDG were located in the green area with coordinates of (*x* = 0.0889, *y* = 0.51747), (*x* = 0.0887, *y* = 0.52913), and (*x* = 0.0950, *y* = 0.52027), respectively. The inset shows the luminescence of the three samples under a 365nm UV lamp, which corresponds to their CIE diagram.

### 3.5. Anti-Counterfeiting Application

In order to further investigate the anti-counterfeiting performance of fluorescent powders, we mixed the three prepared fluorescent powders with ink to form fluorescent ink. In this study, we used screen-printing technology to print the fluorescent ink, which we prepared in the four-leaf-clover pattern. Figure 11a–c show the four-leaf-clover images printed using SAOE fluorescent ink, SAOED fluorescent ink, and SAOEDG fluorescent ink, respectively. It can be clearly observed that nothing could be seen in the day light. When illuminated with a 365nm ultraviolet lamp, it can be seen that all three types of fluorescent ink after screen printing emitted a clear green color. After removing the excitation of the UV lamp, the green luminescence of the SAOE fluorescent ink disappears. However, for SAOED fluorescent ink and SAOEDG fluorescent ink, they still emitted a relatively clear green light. This is related to their long afterglow properties. In addition, fluorescent inks are low-cost and easy to use. Therefore, they can be well applied in the field of anti-counterfeiting.

## 4. Conclusions

In summary, a series of SAOE, SAOED, and SAOEDG phosphors were prepared via a combustion method. XRD, XPS, SEM, and EDX analyses showed that Eu^2+^, Dy^3+^ and Gd^3+^ were successfully doped into a SrAl_2_O_4_ lattice, and the chemical valence and composition of dopant ions were consistent with expectations. These samples all emitted green light under 508 nm ultraviolet excitation, which was ascribed to the 4f^6^5d^1^ to 4f^7^ transition of Eu^2+^. The optimum samples were determined by temperature and concentration control experiments, and their luminescent properties were compared. The results showed that with the addition of Dy and Gd ions, the fluorescence lifetime, luminous intensity, and afterglow time gradually increased, but the quantum yield decreased. The longest fluorescence lifetime and afterglow time of SAOEDG were 839 ns and 54 s. The highest quantum yield of SAOE was 65.87%. The bandgap values calculated through experiments and theoretical calculations achieved the expected results. The prepared SAOE, SAOED, and SAOEDG phosphors were successfully combined with ink and applied to screen printing, which produced a good anti-counterfeiting effect.

## Figures and Tables

**Figure 1 nanomaterials-13-02034-f001:**
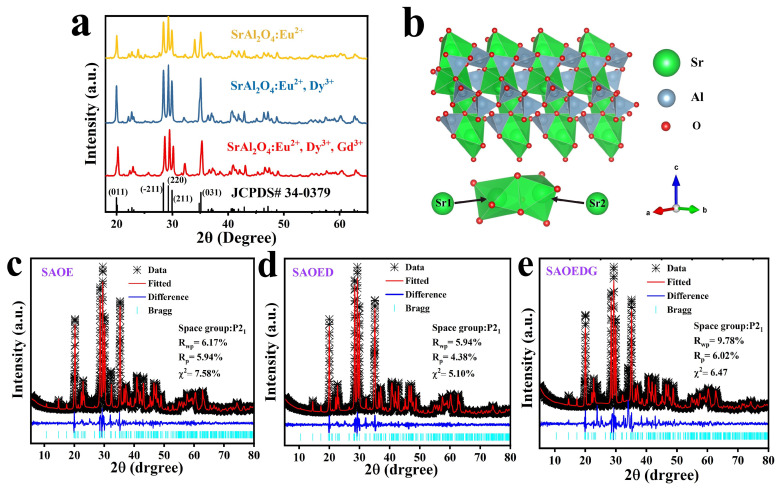
(**a**) X-ray diffraction (XRD) patterns of SrAl_2_O_4_ phosphors doped with different ions, (**b**) crystal structure and polyhedral coordination environment of SrAl_2_O_4_, (**c**) Rietveld refinement of the SAOE phosphor, (**d**) Rietveld refinement of the SAOED phosphor, and (**e**) Rietveld refinement of the SAOEDG phosphor.

**Figure 2 nanomaterials-13-02034-f002:**
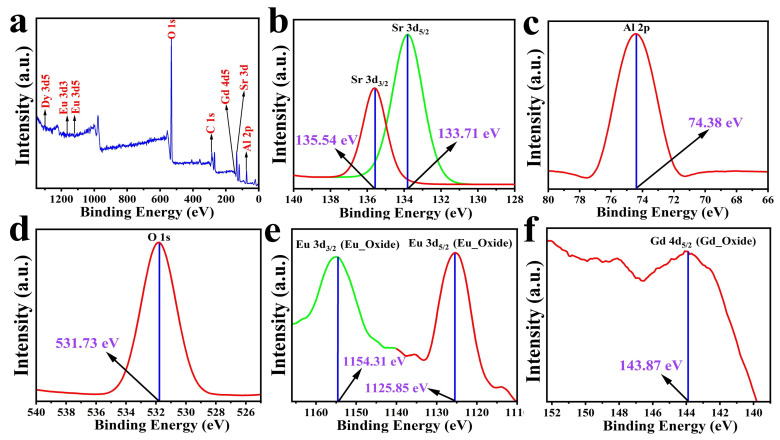
(**a**) The XPS spectra of SrAl_2_O_4_: Eu^2+^, Dy^3+^, Gd^3+^ phosphor, (**b**) the high-resolution XPS spectra of Sr 3d, (**c**) the high-resolution XPS spectra of Al 2p, (**d**) the high-resolution XPS spectra of O 1s, (**e**) the high-resolution XPS spectra of Eu 3d, and (**f**) the high-resolution XPS spectra of Gd 4d.

**Figure 3 nanomaterials-13-02034-f003:**
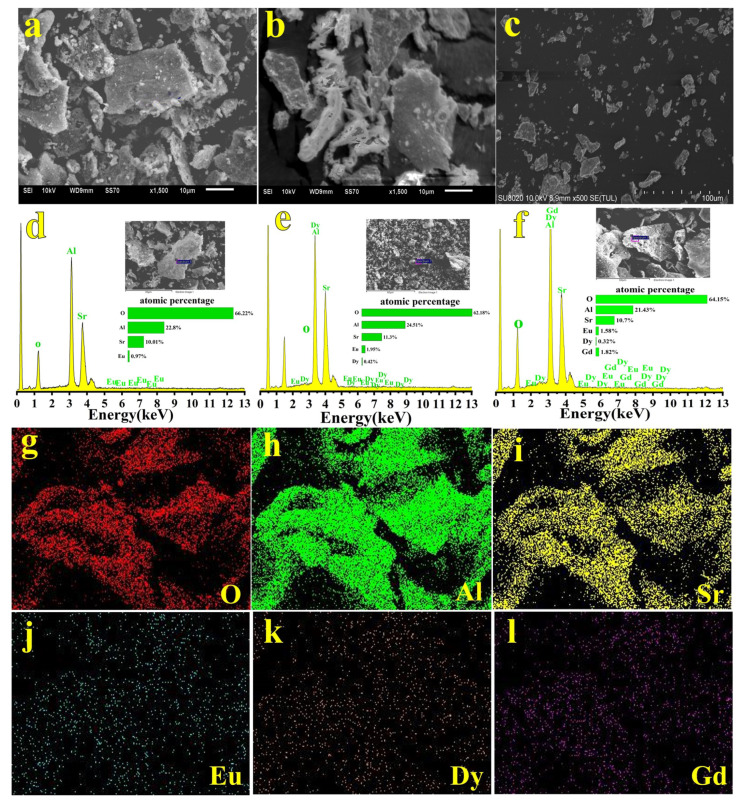
(**a**–**c**) Scanning electron microscope (SEM) images of SAOE, SAOED, and SAOEDG, respectively. (**d**–**f**) The energy dispersive X-ray spectroscopy (EDX) spectra of SAOE, SAOED, and SAOEDG, respectively. (**g**–**l**) Elemental mapping images of O, Al, Sr, Eu, Dy, and Gd, respectively.

**Figure 4 nanomaterials-13-02034-f004:**
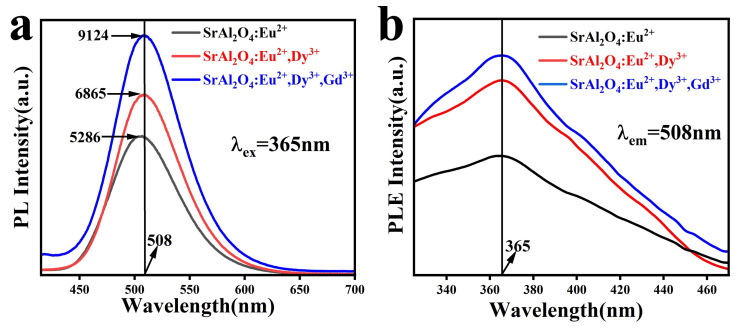
(**a**) Photoluminescence spectroscopy (PL) emission spectrograms of SAOE, SAOED, and SAOEDG samples excited at 365 nm; (**b**) photoluminescence excitation (PLE) excitation spectra of SAOE, SAOED, and SAOEDG samples detected at 508 nm.

**Figure 5 nanomaterials-13-02034-f005:**
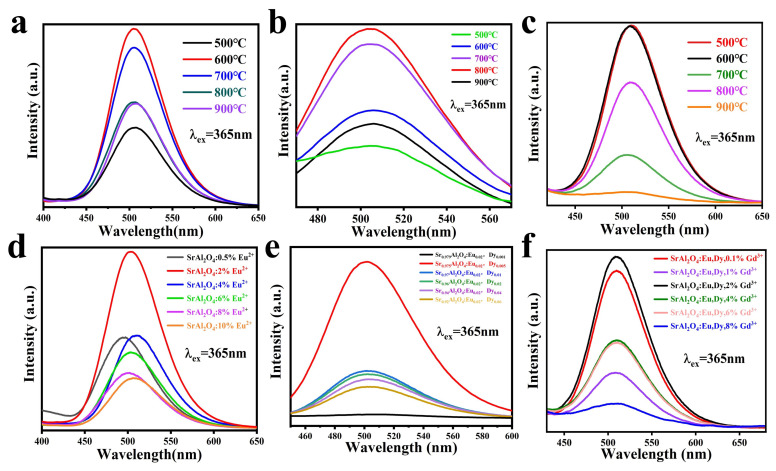
(**a**–**c**) PL spectra of SAOE, SAOED, and SAOEDG samples excited at 365 nm at different temperatures; (**d**–**f**) PL spectra of SAOE, SAOED, and SAOEDG samples excited at 365 nm at different ion concentrations.

**Figure 6 nanomaterials-13-02034-f006:**
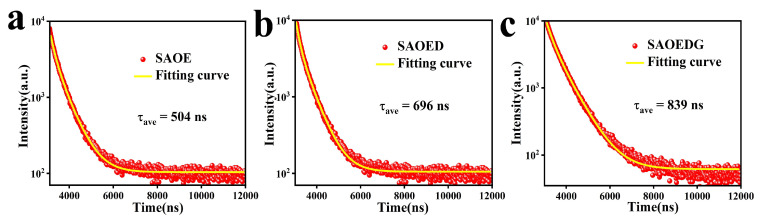
Fluorescence lifetime decay curve and fitting curve of (**a**) SAOE, (**b**) SAOED, and (**c**) SAOEDG.

**Figure 7 nanomaterials-13-02034-f007:**
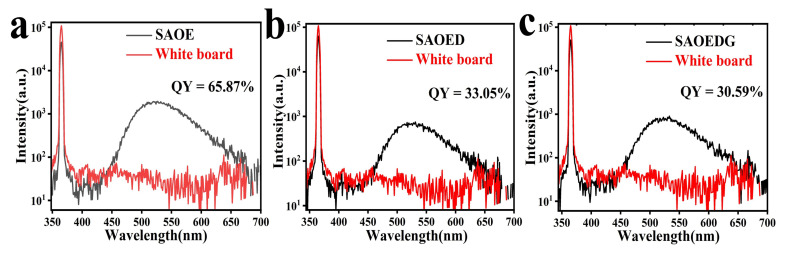
The quantum yields of (**a**) SAOE, (**b**) SAOED, and (**c**) SAOEDG.

**Figure 8 nanomaterials-13-02034-f008:**
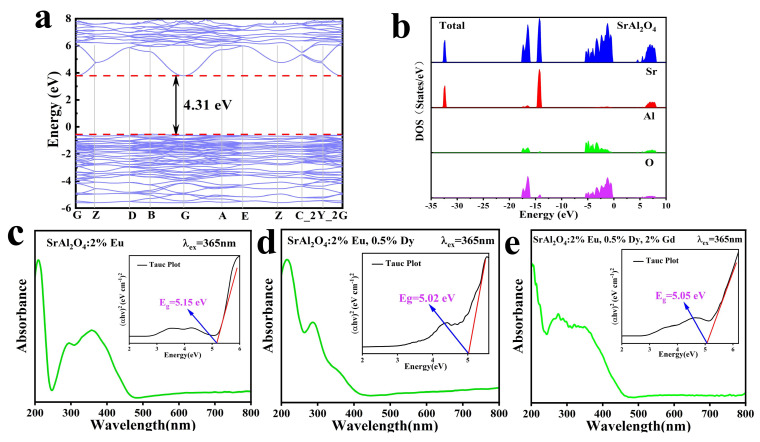
(**a**) Electronic band structure of SrAl_2_O_4_ calculated with DFT; (**b**) total DOS and partial DOS of SrAl_2_O_4_. UV absorption diffuse reflection spectrum of (**c**) SAOE, (**d**) SAOED, and (**e**) SAOEDG (inset: the bandgap values).

**Figure 9 nanomaterials-13-02034-f009:**
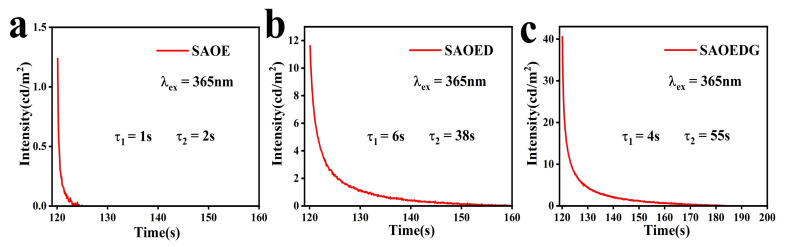
Afterglow decay curves of (**a**) SAOE, (**b**) SAOED, and (**c**) SAOEDG.

**Figure 10 nanomaterials-13-02034-f010:**
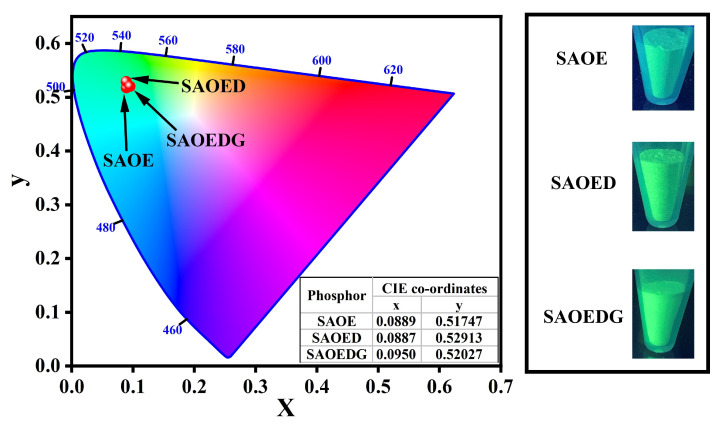
CIE (1976) chromaticity coordinates diagram of SAOE, SAOED, and SAOEDG phosphors under excitation from 365 nm ultraviolet light (inset: luminescence of three samples under 365 nm UV lamp).

**Figure 11 nanomaterials-13-02034-f011:**
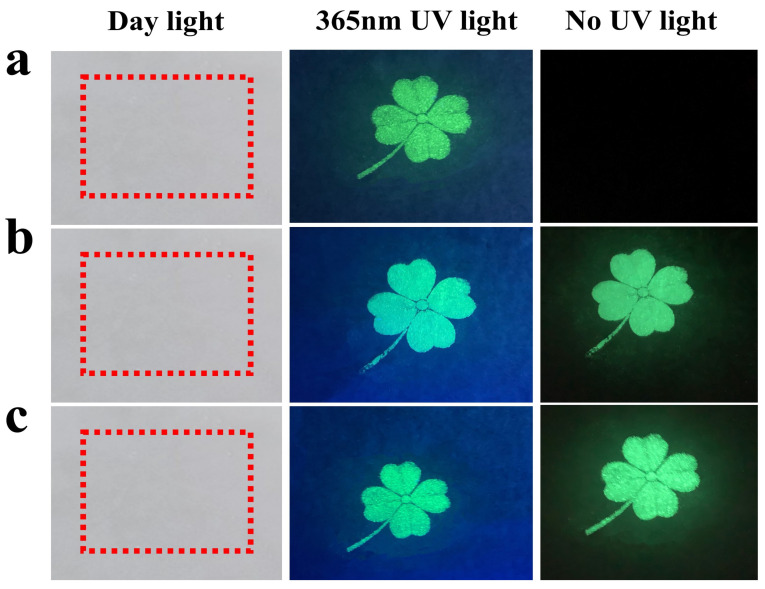
Four-leaf-clover screen-printed pattern for different phosphors: (**a**) SAOE, (**b**) SAOED, and (**c**) SAOEDG.

**Table 1 nanomaterials-13-02034-t001:** XRD data of Sr_0.98_Al_2_O_4_: Eu_0.02_ phosphor.

2θ	FWHM (β)	Lattice Spacing (d)	Intensity (I)	hkl	Crystallite Size (D)
19.951	0.298	4.447	52	011	27.068
28.386	0.286	3.142	100	−211	28.652
29.275	0.257	3.048	91	220	31.949
29.922	0.276	2.984	76	211	29.794
35.113	0.350	2.554	69	031	23.807

**Table 2 nanomaterials-13-02034-t002:** XRD data of Sr_0.975_Al_2_O_4_: Eu_0.02_, Dy_0.005_ phosphor.

2θ	FWHM (β)	Lattice Spacing (d)	Intensity (I)	hkl	Crystallite Size (D)
19.951	0.231	4.447	52	011	34.858
28.386	0.256	3.142	100	−211	31.944
29.275	0.235	3.048	91	220	34.890
29.922	0.251	2.984	76	211	32.826
35.113	0.362	2.554	69	031	23.048

**Table 3 nanomaterials-13-02034-t003:** XRD data of Sr_0.955_Al_2_O_4_: Eu_0.02_, Dy_0.005_, Gd_0.02_ phosphor.

2θ	FWHM (β)	Lattice Spacing (d)	Intensity (I)	hkl	Crystallite Size (D)
19.951	0.222	4.447	52	011	36.285
28.386	0.265	3.142	100	−211	30.899
29.275	0.279	3.048	91	220	29.467
29.922	0.308	2.984	76	211	26.709
35.113	0.320	2.554	69	031	26.074

**Table 4 nanomaterials-13-02034-t004:** Parameters of fluorescence lifetime decay curves for SAOE, SAOED, and SAOEDG samples.

Sample	Decay Lifetime (ns)					
	$A_{1}$	$A_{2}$	$\tau_{1}$	$\tau_{2}$	$\tau_{\mathrm{ave}}$	$\chi^{2}$
SAOE	4828	2733	250	672	504	1.203
SAOED	7321	1248	467	1214	696	1.238
SAOEDG	3794	1179	299	1254	839	1.260

**Table 5 nanomaterials-13-02034-t005:** Afterglow decay time for SAOE, SAOED, and SAOEDG samples.

Sample	Afterglow Decay Time (s)				
	$A_{1}$	$A_{2}$	$\tau_{1}$	$\tau_{2}$	$\tau_{\mathrm{ave}}$
SAOE	3	1	1	2	1
SAOED	4	337	6	38	38
SAOEDG	5	37	4	55	55

## Data Availability

Research data are not available.

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
