# Peer review of "Enhanced Fluorescence Characteristics of SrAl2O4: Eu2+, Dy3+ Phosphor by Co-Doping Gd3+ and Anti-Counterfeiting Application"

_nanomaterials, 2023, doi:10.3390/nano13142034_

Round 1

Reviewer 1 Report

What is "Search engine marketing (SEM) "?  Perhaps it is good to read the manuscript after the automatic edition. The whole paper should be checked to detect such mistakes.

Reviewer 2 Report

The authors submitted an article on triple-doped SrAl2O4 green phosphor with Eu2+, Dy3+ and Gd3+. The idea and the results are interesting to some extent, but the research execution and presentation does not allow for reaching suggested conclusions. I believe the manuscript requires a major revision before it can be published in nanomaterials. Here is the list of issues to be addressed:

1. The structural integrity of the samples is not demonstrated well enough. From the XRD results, there is an apparition of additional phase (peaks at around 26, 27, 31-32 deg) Also the EDX results should be presented in atomic percentage and compared with material’s formula. In text these results are referred to as atomic percentage, which is also confusing.

2. The novelty of the research is questionable, since there was a study on SrAl2O4:Eu,Dy,Gd prepared by combustion method from 2008, which the authors cited in [21]. Yet, this research is not mentioned in the Introduction and no comparison is made with the results of this study.

4. In Experimental, the ink is produced using glass beads, but there is no specification on what kind of glass it is.

5. The claim about only monoclinic phase of SrAl2O4:Eu exhibit luminescence requires proper citation.

6. Line 164-165: The Sr sites are 6-coordinated.

7. The authors report that no PL peaks from Gd and Dy are reported, yet the PL spectra are measured in spectral range in which the emission from Gd would not be visible. Also, the lack of emission from Gd/Dy has nothing to do with location of Eu at Sr site.

8. Line 250: If Dy and Gd act as sensitizers, there would be additional bands on excitation spectra, which is not the case. Simply enhancing the luminescence of Eu does not make these co-dopants sensitizers. To be sensitizers, they would have to absorb light and transfer it to Eu, all of which should be recorded and analyzed.

9. Line 283-284: the ability to absorb light has nothing to do with fluorescence lifetime.

10. Line 285: The claim about controlling lifetime by changing concentration of Dy/Gd is not demonstrated by experimental data.

11. Line 366: what are “large transfer energies” of Dy/Gd? How are holes transported into conduction band? The whole paragraph about trapping mechanism is incorrect and needs to be rewritten.

12. Fig 9a is unclear. It looks as if the luminescence of SAOE and SAOED never stop.

13. Fig. 10 – how can CIE of the samples be different, if the shape of emission spectrum do not change (Fig. 4a)?

14. Fig. 11 – “no UV light” a) has visibly been taken with different setting than b and c, which does not allow for a good comparison.

3. The manuscript requires severe editing and phrasing check-up: SEM abbreviation is expanded as Search Engine Marketing, there is a reference to “humor technology” which I think is translation error, Fig. 1b is referred to before Fig 1a, in line 181 the “Scheller’s formula” should be “Scherrer’s formula”, some sentences throughout the manuscript are in imperative case or lack verbs.

Reviewer 3 Report

"Z. Li and others presented the synthesis of mixed oxide luminescent substances whose properties vary by the ratio of the doping lanthanide ion. In general, the approach to obtaining inorganic luminophores is well known. The selected matrix SrAl2O4 has also long been investigated in combination of europium, dysprosium, and other lanthanides. Including in the article it is necessary to make reference to these works. The mechanism of energy transfer between ions in an inorganic matrix has also been proposed earlier and should be mentioned.
https://doi.org/10.1016/S0022-2313(97)00012-4
https://doi.org/10.1016/S0022-2313(99)00145-3
https://doi.org/10.1016/j.jallcom.2010.04.186
https://doi.org/10.1016/j.jlumin.2018.03.088
It is also necessary to correct the explanation of the high values of the afterglow based on the mentioned works.
The conclusions postulate "The longest fluorescence lifetime and afterglow time of SAOEDG is 839.206 ns and 54.504 s." Why seconds in the latter case and not ns? You need to round the values to integers, e.g., 839 ns.
Note and suggestion: why are the absorption spectra of substances not given? In their absence, the logic of the study in choosing the excitation length is not clear."

Round 2

Reviewer 2 Report

I appreciate the effort of the authors to improve their manuscript and I see, that certain amount of work has been done. However, I do not think the manuscript is ready to be published yet. It still requires a major revision. Here are my answers to some of authors’ comments:

1.       Line 133 “The matrix of glass beads is made of special glass or plastic” – Could you be more specific? Is it glass or is it plastic? Is it silicate, borosilicate, fluoride glass etc.? What kind of plastic is it?

2.       The claim that “It is generally believed that when SrAl2O4 is doped with rare earth ions, only the monoclinic phase exhibits luminescent characteristics.” Still is not properly supported by citation. How do you know that the high-temperature hexagonal phase does not exhibit luminescence? You could argue, that only for monoclinic SrAl2O4 the luminescence was recorded, but it is a different claim.

3.       Line 173: “In the SrAl2O4 crystal structure, there are only two coordinated Sr2+ sites” – I think the authors missed the point of my previous comment. I want to stress, that the Sr sites in SrAl2O4 are 6-coordinated, not “two coordinated”. Also the substitution of Sr by Eu, Gd or Dy requires only a similar ionic radius. The Sr site can be 6-coordinated or 8-coordinated.

4.       In the reply authors say that: “Normally, Dy and Gd as sensitizers do not cause additional spectral bands in the spectrum.” And “This process does not result in additional spectral bands, it only enhances the luminescent effect of the target fluorescent material.” This is demonstrably untrue. There are studies on sensitization by Gd3+, which report additional bands in the excitation spectrum: DOI: 10.1016/j.optmat.2020.110657, DOI: 10.1016/j.jlumin.2020.117380, DOI:10.1016/j.jssc.2013.02.022. Later the authors write that “their characteristic peaks may not be obvious or weak in the excitation spectrum”, which contradicts the previous statement about the lack of peaks. Please substantiate the claim about sensitization by Gd3+ and Dy3+ by something more than an increase in the Eu2+ emission intensity.

5.       The paragraph about decay time was removed completely and now the results are left without comment.

6.       The paragraph about the trapping mechanism is still incorrect. Dy and Gd are not traps, let alone “ion traps”. They create traps by their mismatched charge (trivalent ion in divalent site). The carriers being trapped are not holes, but electrons and they are released to conduction band, not valence band. Please revise thoroughly this part of the manuscript.

7.       Fig. 9a remained unchanged, despite my comment. The authors provided a description of the figure, but it is still, in my opinion, incorrectly displayed. If SAOEDG’s emission intensity goes to zero, then zero on the graph is lower, than the lowers point for SAOE and SAOED, which mean the emission never stops. Please correct it, so that all the plots go to the same zero value.

8.       The authors wrote in their reply about CIE coordinates: “SAOED phosphors add Dy ions on the basis of Eu, and the luminescent characteristics of Dy ions will affect the overall CIE coordinates”, but they did not observe any Dy peaks. So how did DY affect CIE coordinates? The authors also wrote: “The presence of Gd can enhance the luminous intensity of Eu and Dy and make the green fluorescence brighter, but this will also have a slight impact on the CIE coordinates”. That is incorrect. The intensity of emission does not affect CIE coordinates. The X, Y and Z parameters necessary to calculate (x,y) coordinates are normalized, therefore the intensity of the emission does not matter.

9.       The authors wrote in their reply about Fig. 11 that “The Four-leaf clover patterns printed with three fluorescent inks look almost the same, and are not photographed in different settings from b and c.” The area around the print is grey in Fig. 11a, while in the b and c it is black. They must have been photographed with different setting of aperture or shutter speed. If the setting were the same and “SAOE will stop emitting light when there is no ultraviolet light” then the photograph would be all black.

1.       Some of the editing mistakes have been corrected, but there are still missing verbs in some sentences, the imperative case is still used in the manuscript and the mistranslated “humor technology” is still present. Please revise the manuscript carefully.

Round 3

Reviewer 2 Report

The issues with the manuscript were addressed and the quality of the manuscript improved significantly. I recommend the article to be published in Nanomaterials.